# On the Effectiveness of Poisoning against Unsupervised Domain Adaptation

**Akshay Mehra** [1]  **Bhavya Kailkhura** [2]  **Pin-Yu Chen** [3]  **Jihun Hamm** [1]

## Abstract

Data poisoning attacks manipulate victim's training data to compromise their model performance, after training. Previous works on poisoning have shown the inability of a small amount of poisoned data at significantly reducing the test accuracy of deep neural networks. In this work, we propose an upper bound on the test error induced by additive poisoning, which explains the difficulty of poisoning against deep neural networks. However, the limited effect of poisoning is restricted to the setting where training and test data are from the same distribution. To demonstrate this, we study the effect of poisoning in an unsupervised domain adaptation (UDA) setting where the source and the target domain distributions are different. We propose novel data poisoning attacks that prevent UDA methods from learning a representation that generalizes well on the target domain. Our poisoning attacks significantly lower the target domain accuracy of state-of-the-art UDA methods on popular benchmark UDA tasks, dropping it to almost 0% in some cases, with the addition of only 10% poisoned data. The effectiveness of our attacks in the UDA setting highlights the seriousness of the threat posed by data poisoning and the importance of data curation in machine learning.

## 1. Introduction

Data poisoning (Biggio et al., 2012; Mei & Zhu, 2015; Jagielski et al., 2018; Chen et al., 2017; Ji et al., 2017; Mehra et al., 2020) is a training time attack where the attacker has access to the training data which will be used by the victim for model training. The attacker's goal is to modify the training data such that the victim's model performs as the attacker intended after training. A popular method for data poisoning is additive poisoning where victim's training data is augmented with a small amount of poisoned data with the goal of reducing the test accuracy after model train-

ing. Additive poisoning is very effective when models are trained with simple classifiers such as logistic regression (Muñoz-González et al., 2017; Mei & Zhu, 2015; Mehra & Hamm, 2019). However, its effectiveness against deep neural networks has been limited (Huang et al., 2020; Shafahi et al., 2018; Muñoz-González et al., 2017; Mehra & Hamm, 2019), leading to works where several poisoning points are added to affect the classification of a single test point. In this work, we analyze additive data poisoning and derive an upper bound on the test error it can induce. Our upper bound shows that for flexible classifiers such as deep neural networks which can perfectly fit the poisoned training data, the decrease in test accuracy on the clean data is proportional to the amount of poisoned data added.

However, this difficulty of additive poisoning is a consequence of the assumption that training and test data are sampled from the same distribution (single domain poisoning). For more practical settings such as unsupervised domain adaptation (UDA) where knowledge from a label-rich source domain is transferred to an unlabeled target domain with different data distribution, the effect of poisoning has not been studied. In this work, we propose novel data poisoning attacks against UDA methods that learn a domain invariant representation while minimizing the error on the source domain data. Our attacks which include clean label and mislabeled data as poisoning points lead state-of-the-art UDA methods to learn representations that fail to generalize on the target domain on popular UDA tasks. The presence of poisoned data causes UDA methods to either align incorrect classes from the two domains or prevent correct classes from being very close in the representation space. Both of these lead to the failure of UDA methods at reducing the target domain error. With the addition of just 10% poisoned data, our attacks can reduce the target domain accuracy to almost 0%, showing the extreme vulnerability of UDA methods to the poisoning. This dramatic failure caused by the poisoning shows that algorithms that rely on strong assumptions about the data distribution for their success (such as the assumption made by UDA methods, that alignment of the source and target distributions can be achieved by the alignment of their marginal feature distributions without target domain labels) make these methods vulnerable to poisoning. Moreover, our results underscore the importance of data curation in achieving high performance with machine learning methods.

[1]Tulane University [2]Lawrence Livermore National Laboratory [3]IBM Research. Correspondence to: Akshay Mehra <amehra@tulane.edu>.

*Accepted by the ICML 2021 workshop on A Blessing in Disguise: The Prospects and Perils of Adversarial Machine Learning.* Copyright 2021 by the author(s).

## 2. Related work

**Analysis of UDA:** Previous works (Ben-David et al., 2007; 2010; Mansour et al., 2009; Mansour & Schain, 2014) have studied the problem of learning under the UDA setting, leading to the upper bound on the target domain error which inspired many UDA algorithms. Despite the success of these UDA algorithms on benchmark datasets, some works (Ganin et al., 2016; Liu et al., 2019; Zhao et al., 2019; Wang et al., 2019) have presented evidence of their failure in different scenarios. Recently, (Zhao et al., 2019; Combes et al., 2020) provided a lower bound on target domain error when label distributions differ across the two domains. However, their lower bound does not explain all failure cases of UDA including the failure in presence of poisoned data, since a small fraction of poisoned data does not change the label distributions of the two domains. Thus, our poisoning results warrant further research into failure modes of UDA.

**Algorithms for UDA:** Most UDA algorithms (Ganin et al., 2016; Tzeng et al., 2017; Long et al., 2017; Zhao et al., 2018) learn a domain invariant representation while minimizing source domain error. DANN (Ganin et al., 2016) is a popular approach to UDA that uses a discriminator to identify points from the two domains based on their representations. CDAN (Long et al., 2017) is another approach that combines classifier output and representation to identify the domain of a point. Recently, IW-DAN and IW-CDAN (Combes et al., 2020) were proposed as extensions of the original DANN and CDAN with an importance weighting scheme to minimize the mismatch between the labeling distributions of the two domains. Another approach is MCD (Saito et al., 2018) which uses two task-specific classifiers as discriminators to align the domains. The method adversarially trains the representation to minimize the disagreement of the classifiers on the target domain data (classifier discrepancy) while training the classifiers to maximize this discrepancy. A recent approach, SSL (Xu et al., 2019) uses self-supervised tasks (e.g. rotation angle prediction) to align the domains. In this work, we study the effect of poisoning on these state-of-the-art UDA methods.

## 3. Effectiveness of poisoning in single domain versus UDA setting

**Notation:** $\mathcal{X}$ denotes the data domain and $\mathcal{D}$ is a distribution defined on this domain. Let $f : \mathcal{X} \to [0, 1]$ be a deterministic labeling function, for the given binary classification task which can be interpreted as $Pr[y = 1|x]$ and $h : \mathcal{X} \to [0, 1]$ denote a hypothesis function. Then the probability of disagreement (Ben-David et al., 2010) between two hypotheses functions $h$ and $g$ according to the data distribution $\mathcal{D}$ is given by $\epsilon_{\mathcal{D}}(h, g) := E_{x \sim \mathcal{D}}|h(x) - g(x)| = Pr[h(x) \neq g(x)]$. Let $\hat{\mathcal{D}}$ denote a sample from the data distribution $\mathcal{D}$.

### 3.1. Poisoning in a single domain setting

Here we study the limits of additive data poisoning attacks in a single domain setting at reducing the test accuracy of models trained with empirical risk minimization (ERM). The process of augmenting victim's clean training data $\hat{\mathcal{D}}_{\text{clean}}$ with a fraction $\rho$ of poisoned data $\hat{\mathcal{D}}_{\text{poison}}$ to obtain the poisoned training data $\hat{\mathcal{D}}_{\text{poisoned}}$ (i.e., $\hat{\mathcal{D}}_{\text{poisoned}} = \hat{\mathcal{D}}_{\text{clean}} \cup \hat{\mathcal{D}}_{\text{poison}}$) has the following probabilistic analog.

$$P_{\text{poisoned}}(x) = \frac{1}{1+\rho}P_{\text{clean}}(x) + \frac{\rho}{1+\rho}Q(x),$$

where $0 < \rho < 1$ is the fraction of the poisoned data, $P_{\text{clean}}(x)$ is the clean data distribution, $Q(x)$ is the distribution of poison data and $P_{\text{poisoned}}$ is the data distribution obtained after poisoning. Theorem 1 provides an upper bound on the target error induced by additive poisoning.

**Theorem 1.** *Let $\mathcal{H}$ be the hypothesis class on $\mathcal{X}$. Then $\forall h, g \in \mathcal{H}, \epsilon_{\text{clean}}(h, g) - \epsilon_{\text{poisoned}}(h, g) \leq \rho$.*

The theorem implies that if the disagreement between two hypotheses $h$ and $g$ is minimized on the data distribution after poisoning then the disagreement between these two hypotheses on the clean data distribution is upper bounded by the percentage of poison data $\rho$. Thus, hypotheses that work well on the poisoned data cannot be ill-performing on the clean data, especially in presence of a small fraction of poison data. This upper bound explains the difficulty of additive poisoning at significantly reducing the test accuracy using a small amount of poisoned data, especially when using deep neural networks for classification.

### 3.2. Poisoning in UDA setting

In this section, we present novel data poisoning attacks against UDA methods. Our additive poisoning attacks are designed to fool UDA methods that learn a domain invariant representation while minimizing the error on the source domain into producing a representation that leads to a high error on the target domain data. As in additive poisoning, the poisoned data generated through our attacks will be added to the clean source domain data. The poisoned source domain data along with unlabeled target domain data will then be used to train UDA methods. We now describe our attacks that use mislabeled and watermarked data for poisoning. Due to space limitation, description and experiments of clean label attack are presented in the Appendix B.

**Poisoning with mislabeled source and target domain data.** We propose two types of poisoning attacks that make use of mislabeled data, namely, wrong-label correct-domain attack and wrong-label incorrect-domain attack. In wrong-label correct-domain attack, poison data is selected from the source domain. This data along with incorrect labels is then added to clean source domain data. Since the poison data only has wrong labels the attack is termed wrong-label

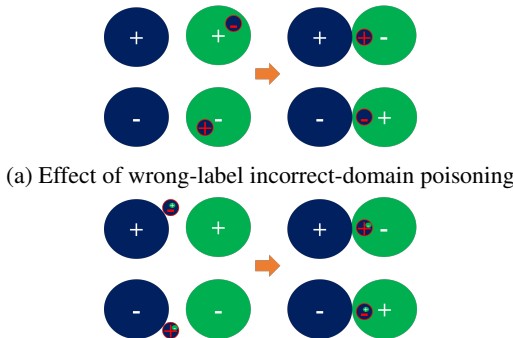

(a) Effect of wrong-label incorrect-domain poisoning

(b) Effect of poisoning with watermarked data

*Figure 1.* Poisoning with mislabeled data causes discriminator-based UDA methods to align wrong classes (+ to -) from the source and target domains.

correct-domain attack. In wrong-label incorrect-domain attack, poison data selected from the target domain along with incorrect labels is used to form the poisoned source domain data. Since the poisoned source domain contains the target data, which does not belong to the source domain, the attack is termed wrong-label incorrect-domain. The failure of UDA methods in learning a representation that increases target error in presence of poisoned data generated through these attacks depends on the labels assigned to the poison data. We describe two simple yet effective methods for choosing labels for the poison data in Sec. 4.

**Poisoning using watermarked data.** In this attack, poisoned data is generated by superimposing an image from the source domain with an image from the target domain with incorrect labels, while ensuring the target image is not visible in the poison data. This makes the poison data looks like the correct source domain data. To generate such watermarked poison data we select an image from the target domain ($t$) and a base image from the source domain ($s$) such that it has the same class as the target domain image and lies closest to it (in the input space). The poisoned image ($p$) is obtained by a convex combination of the base and target images i.e., $p = \alpha t + (1-\alpha)s$ where $\alpha \in [0, 1]$. The value $\alpha$ governs how prominent the target image is in the poison data. The success of poisoning, as with the previous attack, depends on the choice of labels used for poison data.

## 4. Experiments

In this section, we present the results of our data poisoning attacks against state-of-the-art UDA methods. We use DANN (Ganin et al., 2016), CDAN (Long et al., 2017), MCD (Saito et al., 2018), SSL (Xu et al., 2019) (with rotation-angle prediction task), IW-DAN (Combes et al., 2020), and IW-CDAN (Combes et al., 2020) for our experiments. We use two benchmark datasets, namely Digits and Office-31. Under Digits, we evaluate four tasks using SVHN (S), MNIST (M), MNIST_M (MM), and USPS (U) datasets

and six tasks under the Office-31 using Amazon (A), DSLR (D), and Webcam (W) datasets. The effect of poisoning is measured by the difference in the target domain accuracy of these methods on clean versus poisoned data. We provide an intuitive picture of how our poisoning attacks prevent UDA methods from learning a representation that generalizes well on the target domain data in Fig. 1. The figure illustrates how discriminator-based UDA methods align wrong classes from the source and target domains in presence of poisoned data. The incorrect alignment leads to a significant drop in the target domain accuracy obtainable with these methods. We contrast these results with poisoning target domain data in a single domain setting. In this setting, models are trained with ERM using labeled target domain training data along with poisoned data and the effect of poisoning is measured by the drop in the test accuracy on the target domain. Our experimental results show that additive poisoning is significantly more effective in the UDA setting than in the single domain setting. Additional experimental details including model architectures are presented in Appendix C.

**Poisoning using mislabeled source and target data.** As mentioned in Sec. 3.2, the effectiveness of poisoning is dependent on the labels of the poisoned data. For the Digits dataset, poison data is labeled as the class next to their true class (e.g. poison points with true class one are labeled as two, points with true class two are labeled as three, and so on). For the Office-31 dataset, we label the poison point to be in the class closest (in the representation space learned using clean source domain data) in the source domain other than its true class. In this experiment, the attacker is limited to adding only 10% poisoned data with respect to the size of the target domain data. The results for wrong-label correct-domain poisoning are present in rows marked with Poison$_{\text{source}}$ in Table 1 and 2. Poisoning with this approach only causes a minor decrease in target domain accuracy as a large amount of correctly labeled source domain data prevents the poisoned data from severely affecting the performance of UDA methods (similar to single domain poisoning setting). On the other hand, wrong-label incorrect-domain poisoning causes the discriminator-based UDA methods (Ganin et al., 2016; Long et al., 2017) to align wrong classes from the two domains as shown in Fig 1(a), leading to a significant drop in the target domain accuracy. This is due to the domain discriminator being maximally confused when marginal distributions of the source and target domains are aligned. However, marginal distribution alignment does not ensure the alignment of the conditional distributions. Moreover, the objective of achieving low source domain error pushes the representation learner to correctly classify the poison data, thereby placing the poison and source data with the same labels close in the representation space. Since the poison data is mislabeled target domain data, wrong classes from the two domains are aligned. For MCD (Saito et al.,

*Table 1.* Decrease in the target domain accuracy for UDA methods trained on poisoned source domain data (with poisons sampled from source/target domains) compared to accuracy attained with clean data on the Digits tasks (mean±s.d. of 5 trials).

| Method | Data | S→ M | M → MM | M → U | U → M |
|---|---|---|---|---|---|
| Source only | Clean | 72.42±1.44 | 39.05±2.30 | 87.13±1.75 | 78.6±1.45 |
| DANN | Clean | 78.05±1.15 | 76.22±2.38 | 92.17±0.73 | 92.73±0.71 |
|  | $\text{Poison}_{\text{source}}$ | 70.26±2.84 | 69.98±3.49 | 93.44±0.84 | 92.08±0.68 |
|  | $\text{Poison}_{\text{target}}$ | **1.46±1.12** | **0.48±0.04** | **0.97±0.53** | **5.83±0.82** |
| CDAN | Clean | 79.19±0.70 | 73.88±1.10 | 93.92±0.97 | 95.94±0.71 |
|  | $\text{Poison}_{\text{source}}$ | 73.67±4.19 | 73.36±1.31 | 92.06±0.59 | 92.85±0.31 |
|  | $\text{Poison}_{\text{target}}$ | **12.27±5.02** | **0.59±0.12** | **1.92±0.42** | **2.96±0.71** |
| MCD | Clean | 96.18±1.53 | 93.95±0.33 | 89.96±2.04 | 88.34±2.50 |
|  | $\text{Poison}_{\text{source}}$ | 85.86±5.66 | 93.33±0.71 | 87.99±1.05 | 83.19±2.98 |
|  | $\text{Poison}_{\text{target}}$ | **0.97±0.94** | **0.37±0.06** | **0.66±0.16** | **2.07±0.69** |
| SSL | Clean | 66.85±2.30 | 92.76±0.91 | 88.69±1.28 | 82.23±1.59 |
|  | $\text{Poison}_{\text{source}}$ | 61.97±1.62 | 91.35±1.13 | 85.74±2.92 | 82.56±0.84 |
|  | $\text{Poison}_{\text{target}}$ | **0.31±0.03** | **0.36±0.02** | **7.76±1.52** | **9.88±1.07** |

*Table 2.* Decrease in the target domain accuracy for UDA methods trained on poisoned source domain data (with poisons sampled from source/target domains) compared to accuracy attained with clean data on the Office-31 tasks (mean±s.d. of 3 trials).

| Method | Dataset | A → D | A → W | D → A | D → W | W → A | W → D |
|---|---|---|---|---|---|---|---|
| Source only | Clean | 79.61 | 73.18 | 59.33 | 96.31 | 58.75 | 99.68 |
| DANN | Clean | 84.06 | 85.41 | 64.67 | 96.08 | 66.77 | 99.44 |
|  | $\text{Poison}_{\text{source}}$ | 79.11±0.35 | 83.98±1.19 | 44.31±2.94 | 95.22±0.22 | 43.35±1.65 | 96.58±0.87 |
|  | $\text{Poison}_{\text{target}}$ | **59.83±0.20** | **63.18±1.96** | **17.58±0.39** | **76.43±0.62** | **19.82±0.33** | **84.20±0.71** |
| CDAN | Clean | 89.56 | 93.01 | 71.25 | 99.24 | 70.32 | 100 |
|  | $\text{Poison}_{\text{source}}$ | 90.16±0.61 | 90.94±0.13 | 53.68±0.37 | 98.45±0.07 | 57.27±0.57 | 99.66±0.23 |
|  | $\text{Poison}_{\text{target}}$ | **71.88±0.20** | **71.94±0.76** | **11.19±1.47** | **86.37±0.36** | **18.54±0.45** | **89.08±1.23** |
| IW-DAN | Clean | 84.3 | 86.42 | 68.38 | 97.13 | 67.16 | 100 |
|  | $\text{Poison}_{\text{source}}$ | 81.25±0.91 | 83.27±0.45 | 50.76±1.58 | 96.68±0.29 | 48.31±2.02 | 99.73±0.12 |
|  | $\text{Poison}_{\text{target}}$ | **61.64±0.53** | **63.43±1.14** | **15.69±1.76** | **80.29±0.07** | **26.54±0.48** | **88.62±0.23** |
| IW-CDAN | Clean | 88.91 | 93.23 | 71.9 | 99.3 | 70.43 | 100 |
|  | $\text{Poison}_{\text{source}}$ | 89.83±0.31 | 90.77±1.27 | 57.51±0.06 | 98.41±0.07 | 61.16±1.21 | 99.66±0.12 |
|  | $\text{Poison}_{\text{target}}$ | **72.62±0.42** | **70.15±2.21** | **14.36±0.66** | **88.26±0.15** | **22.36±0.96** | **87.55±0.53** |

2018), which uses use classifier discrepancy to detect and align source and target domains, our poisoned data prevents it from detecting target examples. This is because the term that minimizes the error on the poisoned source domain also reduces the discrepancy of the classifiers on poison data, which are from the target domain. Thus, both the representation learner and discriminator (in the form of two classifiers) become optimal without aligning the two domains. In SSL (Xu et al., 2019), the representation must correctly classify source domain data (main task) including the poisoned data. Although the auxiliary task ensures that representations of the source and target domains become similar but in presence of poisoned data, similar representations of source and target domain classes lead to a drop in the accuracy of the main task on the poisoned data. This creates a conflict between the main and auxiliary tasks, due to which correct source and target domain classes cannot be aligned. The t-SNE plots for these are shown in Fig. 3 and 4 in the Appendix. The results of wrong-label incorrect-domain poisoning, in rows marked with $\text{Poison}_{\text{target}}$ in Tables 1 and 2, show a significant reduction in the target domain accuracy compared to the accuracy obtained on clean data. This success of poisoning in the UDA setting is contrasted with its performance in a single domain setting by adding 10% poison data in the training set of the target domain. The attacks decrease the test accuracy of the target domain by roughly 3% on Digits datasets (As Office-31 does not have

*Table 3.* Decrease in target accuracy when training different domain adaptation methods on poisoned watermarked data in comparison to the target accuracy obtained with clean data on the Digits task (mean±s.d. of 5 trials).

| Method | Dataset | S → M | M → MM | M → U | U → M |
|---|---|---|---|---|---|
| DANN | Clean | 78.05±1.15 | 76.22±2.38 | 92.17±0.73 | 92.73±0.71 |
|  | $\text{Poison}_{\alpha}$ | $68.76\pm3.91_{0.05}$ | $27.36\pm15.77_{0.05}$ | $91.84\pm0.55_{0.10}$ | $88.93\pm4.36_{0.10}$ |
|  |  | $57.96\pm5.84_{0.10}$ | $7.19\pm2.59_{0.10}$ | $85.51\pm3.01_{0.20}$ | $78.29\pm8.52_{0.20}$ |
|  |  | $33.33\pm4.38_{0.15}$ | $4.73\pm0.38_{0.15}$ | $39.29\pm1.34_{0.30}$ | $41.52\pm7.43_{0.30}$ |
| CDAN | Clean | 79.19±0.70 | 73.88±1.10 | 93.92±0.97 | 95.94±0.71 |
|  | $\text{Poison}_{\alpha}$ | $65.77\pm4.82_{0.05}$ | $55.47\pm3.87_{0.05}$ | $92.05\pm0.96_{0.10}$ | $86.53\pm1.55_{0.10}$ |
|  |  | $57.57\pm3.11_{0.10}$ | $7.37\pm1.26_{0.10}$ | $86.54\pm2.43_{0.20}$ | $77.39\pm4.84_{0.20}$ |
|  |  | $44.83\pm4.09_{0.15}$ | $6.68\pm1.64_{0.15}$ | $88.67\pm0.44_{0.30}$ | $79.54\pm7.02_{0.30}$ |
| MCD | Clean | 96.18±1.53 | 93.95±0.33 | 89.96±2.04 | 88.34±2.50 |
|  | $\text{Poison}_{\alpha}$ | $74.96\pm3.20_{0.05}$ | $92.18\pm0.78_{0.05}$ | $6.75\pm4.81_{0.10}$ | $30.35\pm2.30_{0.10}$ |
|  |  | $35.85\pm3.23_{0.10}$ | $85.38\pm3.57_{0.10}$ | $0.77\pm0.22_{0.20}$ | $11.34\pm0.77_{0.20}$ |
|  |  | $17.01\pm1.52_{0.15}$ | $70.34\pm11.49_{0.15}$ | $0.71\pm0.22_{0.30}$ | $3.28\pm0.94_{0.30}$ |
| SSL | Clean | 66.85±2.30 | 92.76±0.91 | 88.69±1.28 | 82.23±1.59 |
|  | $\text{Poison}_{\alpha}$ | $44.64\pm2.01_{0.05}$ | $53.33\pm13.48_{0.05}$ | $32.38\pm10.77_{0.10}$ | $34.72\pm1.71_{0.10}$ |
|  |  | $10.86\pm1.21_{0.10}$ | $26.64\pm10.1_{0.10}$ | $6.12\pm2.13_{0.20}$ | $21.86\pm1.01_{0.20}$ |
|  |  | $3.4\pm1.11_{0.15}$ | $12.14\pm4.66_{0.15}$ | $2.42\pm0.41_{0.30}$ | $11.90\pm0.81_{0.30}$ |

separate test data, we omit single domain poisoning on it). The result shows the limited impact of poisoning in a single domain setting as suggested by our Theorem 1.

**Poisoning using watermarked data.** For this attack, we use the labeling scheme described previously for Digits to assign labels to poisoned data and use 10% as poison percentage. The illustrative picture of the effect of poisoning in this scenario is presented in Fig. 1(b). The figure shows that successful watermarking attacks have the same effect as wrong-label incorrect-domain poisoning attacks, even though the poison data looks like the source domain data. We use different values of $\alpha$ to evaluate the effectiveness of this attack on the Digits dataset. The results in Table 3 show a significant decrease in the target domain accuracy even with a small $\alpha$ for all methods except CDAN. This is because the success of CDAN depends on the correctness of the pseudo-labels on the target domain data (output of the classifier), which are used in the discriminator. Correct pseudo-labels provide CDAN a positive reinforcement to align correct classes from the two domains. However, as the amount of watermarking increases, the quality of pseudo labels deteriorates. Thus, providing a negative reinforcement to CDAN which leads to the alignment of wrong classes across the two domains causing a high target domain error.

## 5. Conclusion

In this work, we studied the effectiveness of poisoning in a single domain and UDA settings. We derived an upper bound on the test error induced by poisoning in a single domain setting explaining the limited effect of poisoning against deep neural networks as observed in previous works. To highlight that the difficulty of poisoning in a single domain setting does not undermine the threat posed by poisoning to machine learning algorithms, we proposed novel poisoning attacks in the UDA setting. The failure of popular UDA methods in presence of the small amount of poisoned data shows the importance of training-data quality and data curation for the success of machine learning methods.

## 6. Acknowledgement

This work was supported by the NSF EPSCoR-Louisiana Materials Design Alliance (LAMDA) program #OIA-1946231 and by LLNL Laboratory Directed Research and Development project 20-ER-014. This work was performed under the auspices of the U.S. Department of Energy by the Lawrence Livermore National Laboratory under Contract No. DE-AC52-07NA27344, Lawrence Livermore National Security, LLC [1].

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

# Appendix

We present the proof of Theorem 1 in App. A followed the clean-label poisoning attack against UDA methods in App. B. We conclude in App. C by providing the details of the datasets and model architectures used in our experiments.

## A. Proofs

**Theorem 1.** $\forall h, g \in \mathcal{H}, \epsilon_{\text{clean}}(h, g) - \epsilon_{\text{poisoned}}(h, g) \leq \rho$.

*Proof.* As $P_{\text{clean}}(x) = (1 + \rho)P_{\text{poisoned}}(x) - \rho Q(x)$, we have,

$$\epsilon_{\text{clean}}(h, g) - \epsilon_{\text{poisoned}}(h, g)$$
$$= \int |h - g|(P_{\text{clean}}(x) - P_{\text{poisoned}}(x)) \, dx$$
$$= \int |h - g|\rho(P_{\text{poisoned}}(x) - Q(x)) \, dx$$
$$\leq \rho \int |h - g|P_{\text{poisoned}}(x) \, dx \leq \rho.$$

$\square$

## B. Poisoning using clean-label data

To generate clean-label poison data that can affect the performance of UDA methods we require solving a bilevel optimization problem, similar to previous works (Huang et al., 2020; Mehra & Hamm, 2019; Mehra et al., 2020). Due to the high computational complexity involved in solving the bilevel problem, we propose an alternating optimization-based variant for generating poisoned data and demonstrate the feasibility of clean-label attacks against UDA with it. Similar to previous works (Huang et al., 2020; Shafahi et al., 2018), we consider misclassification of a single target domain test point $(x_{\text{test}}^{\text{target}}, y_{\text{test}}^{\text{target}})$. Let $u = \{u_1, ..., u_n\}$ denote the poisoned data. To ensure clean label, each poison point $u_i$ must have a bounded perturbation from a base point $x_i^{\text{base}}$ i.e, $\|u_i - x_i^{\text{base}}\| = \|\delta_i\| \leq \epsilon$ and has label of the base i.e., $y_i^{\text{base}}$. Thus, $\hat{\mathcal{D}}^{\text{poison}} = \{(u_i, y_i^{\text{base}})\}_{i=1}^{N_{\text{poison}}}$, $\hat{\mathcal{D}}_{\text{source}} = \{(x_i^{\text{source}}, y_i^{\text{source}})\}_{i=1}^{N_{\text{source}}}$ and $\hat{\mathcal{D}}_{\text{target}} = \{(x_i^{\text{target}}, y_i^{\text{target}})\}_{i=1}^{N_{\text{target}}}$. The clean-label poison data $u$ is such that when the victim uses $\hat{\mathcal{D}}^{\text{source}} \bigcup \hat{\mathcal{D}}^{\text{poison}}$ and $\hat{\mathcal{D}}^{\text{target}}$ (without labels) for UDA, the target domain test point $(x_{\text{test}}^{\text{target}}, y_{\text{test}}^{\text{target}})$ is misclassified. The optimization problem for the clean-label attack is as follows.

$$\min_u \sum_{i=1}^{N_{\text{poison}}} \left[ \|g(x_{\text{test}}^{\text{target}}; \theta) - g(u_i; \theta)\|_2^2 + \lambda \|x_i^{\text{base}} - u_i\| \right],$$
$$\min_\theta \mathcal{L}_{\text{UDA}}(\hat{\mathcal{D}}^{\text{source}} \bigcup \hat{\mathcal{D}}^{\text{poison}}, \hat{\mathcal{D}}^{\text{target}}; \theta),$$
$$(1)$$

where $g$ denotes the representation space embedding of the data. The first problem minimizes the distance between the representations of the poison and the target domain test data (first term) while ensuring the poison data is not too far from the base data (second term). The second problem optimizes the parameters of the representation using UDA methods. Based on the choice of the domain of the base data we propose two clean-label poisoning attacks. The first being a clean-label correct-domain attack in which the base data is picked from the source domain and the second being a clean-label incorrect-domain attack in which base data is chosen from the target domain. Attack success is evaluated by solving the second problem in Eq. 1 from scratch and evaluating the classification of $x_{\text{test}}^{\text{target}}$. Effect of these poisoning attacks on UDA methods is presented in Fig. 2(a). Left part of Fig. 2(a) shows the case before retraining using the poison data generated from Eq. 1 and the right part shows how poisoning induces misclassification. For this experiment, 1% poison data is used to prevent the alignment of a target test point to its correct class. We test the attack on binary classification problem (3 vs 8) on MNIST→ MNIST_M. We initialize the poison data from the class opposite to the true class of the target test point and using the points closest (in the input space) to the target

test point. The poison data obtained after solving Eq. 1 is added to the source domain data and UDA methods are retrained from scratch. The attack is considered successful if the target test point is misclassified after this retraining. For the results shown in Fig. 2(b), we randomly targeted 20 points and obtained poison data corresponding to each UDA method. Attack success is reported after evaluating UDA methods on five random initializations by adding the generated poison data in the source domain. To control the amount of maximum distortion between experiments, we add a constraint on the maximum permissible distortion to poison data using $\ell_\infty$ norm and use a value of $\epsilon = 0.1$. To generate poison data that remains effective even after UDA methods are trained from scratch, we make use of multiple randomly initialized networks during poison generation. Following the work (Huang et al., 2020), we reinitialize the models at different points during optimization. This re-initialization scheme helps train UDA methods with different random initializations and for a different number of epochs making the poison data more resilient to initialization change that can happen at test-time. The results in Fig. 2(b) show that using target domain data as base data is significantly more successful under small permissible perturbation ($\epsilon$). Using base data from the source domain requires large initial distortion to keep the poison data close to the target point in the representation space and is hence less successful.

For single domain poisoning, we add poisoned data to the labeled training set of MNIST_M and consider the classification of a single test point from MNIST_M. We use the same poisoning percentage and maximum distortion as in the UDA poisoning experiments. Unlike the UDA setting where attack success is evaluated on models trained with UDA methods, for single domain setting models are trained with ERM. The average attack success rate for poisoning 20 randomly chosen test points is 58% in this single domain setting. This is lower than the attack success we get with clean-label incorrect-domain poisoning with most UDA methods suggesting the difficulty of poisoning in a single domain setting.

## C. Details of the experiments

All codes are written in Python using Tensorflow/Keras and were run on Intel Xeon(R) W-2123 CPU with 64 GB of RAM and dual NVIDIA TITAN RTX. Dataset details and model architectures used are described below.

### C.1. Dataset description

Here we describe the details of the datasets used for the Digits and Office-31 tasks.

**Digits:** For this task, we use 4 datasets: MNIST, MNIST_M, SVHN, and USPS. We evaluate four

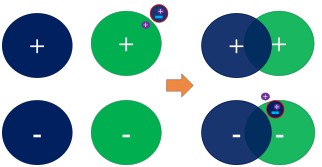

(a) Clean-label correct-domain poisoning attack aligns the target domain test point (purple with label +) close to the wrong class (-).

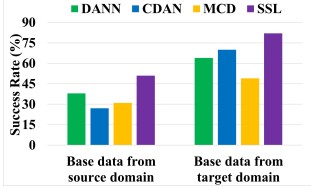

(b) The attack success rate of clean-label poisoning using base data from source/target for a two-class problem in MNIST → MNIST_M.

*Figure 2.* Effect of clean-label poisoning attack

popular tasks under this, namely, SVHN→ MNIST, MNIST→ MNIST_M, MNIST→ USPS and USPS→ MNIST. For SVHN→ MNIST, we train on 73,257 images from SVHN and 60,000 images from MNIST while testing on 10,000 MNIST images. For MNIST→ MNIST_M, we use 60,000 images from MNIST and MNIST_M for training and test on 10,000 MNIST images. Lastly, for MNIST→ USPS and USPS→ MNIST, we use 2,000 images from MNIST and 1,800 images from USPS for training. We test on the 10,000 MNIST images and 1,860 USPS images.

**Office-31:** The dataset contains a total of 4110 images belonging to 31 categories from 3 domains: Amazon (A), DSLR(D), and Webcam(W). We evaluate the performance of UDA on all six tasks, namely, A→ D, A→ W, D→ A, D→ W, W→ A, W→ D.

### C.2. Model architecture

Here we describe the model architectures used for different tasks. To fairly compare the performance of different UDA methods and eliminate the effect of architecture changes in improving the performance of different methods, we make use of similar model architectures for different methods, as described below. The effectiveness of these architectures has also been shown by previous works.

**Digits:** The architectures used for MNIST→ MNIST_M, MNIST→ USPS and USPS→ MNIST involves a shared convolution neural network. The output of this shared network is fed into a softmax classifier and the discriminator. The architecture of the shared network consists of a convolution layer with a kernel size of 5x5, 20 filters, and ReLU activation, followed by a max-pooling layer of size 2x2. This is followed by another convolution layer with a 5x5

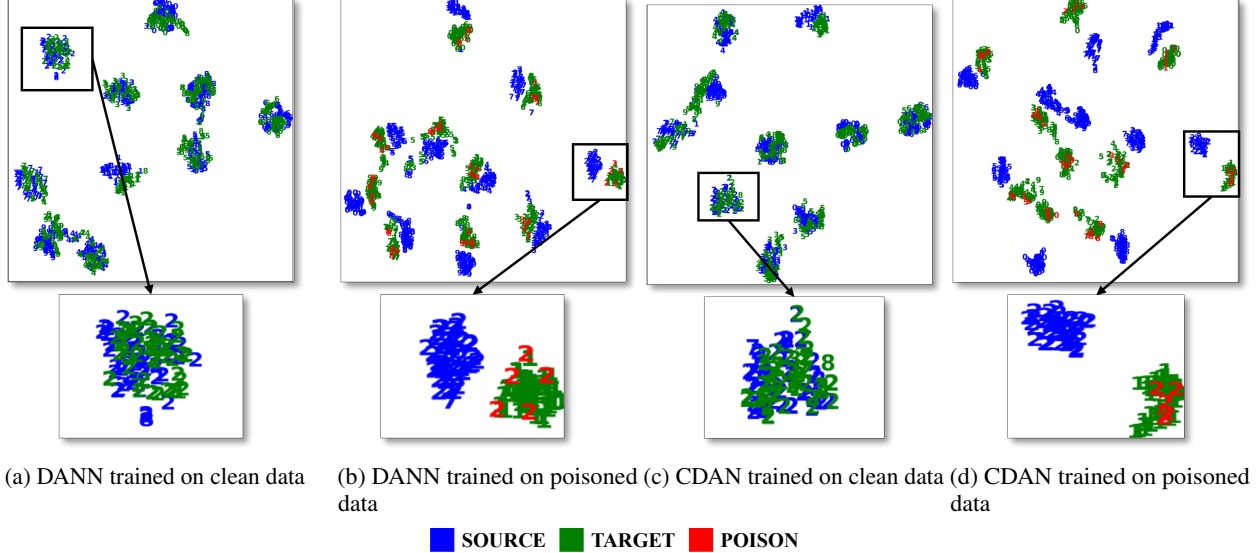

(a) DANN trained on clean data    (b) DANN trained on poisoned (c) CDAN trained on clean data (d) CDAN trained on poisoned
data                                                             data

**■ SOURCE   ■ TARGET   ■ POISON**

*Figure 3.* (Best viewed in color.) t-SNE embedding of the data in the representation space (for MNIST→ USPS task) learned using DANN and CDAN on clean and poisoned source domain data. Without poisoning, correct classes (data from source class 2 is zoomed in) from two domains are aligned ((a) and (c)). The presence of poisoned data fools the methods into aligning incorrect classes from the two domains ((b) and (d)). The mismatch between the source and target classes is dependent on the labels of the poison data (due to which, target class 1 aligned to source class 2).

kernel, 50 filters, and ReLU activation followed by similar max pooling and a dropout. Then we have a fully connected layer with ReLU activation of size 500 followed by a dropout layer. For the discriminator, we use two dense layers with 500 units each followed by a ReLU and a dropout layer. This is followed by a 2 unit softmax layer. For MCD, we use the following architecture for the generator on MNIST→ MNIST_M task. A convolution layer with a kernel size of 5x5, 32 filters, and ReLU activation, followed by a max-pooling layer of size 2x2. This is followed by another convolution layer with a 5x5 kernel, 48 filters, and ReLU activation followed by a similar max-pooling layer. For the classifier, we use 2 dense layers with 100 units followed by ReLU activation and dropout layers. This is followed by the softmax layer. Unlike the original work MCD (Saito et al., 2018), we do not use batch normalization layers in these tasks to make architectures consistent across different methods. For SVHN→ MNIST we use the following architecture for the generator. A convolution layer with a kernel size of 5x5, 64 filters, a stride of 2 followed by batch normalization, dropout, and ReLU activation layer. This is followed by another convolution layer with a kernel size of 5x5, 128 filters, a stride of 2 followed by batch normalization, dropout, and ReLU activation layer. Then another convolution layer with a kernel size of 5x5, 256 filters, a stride of 2 followed by batch normalization, dropout, and ReLU activation layer. This is followed by a dense layer with 512 units followed by batch normalization, ReLU activation, and a dropout layer. We use the softmax layer for classification. For the discriminator, we use two

dense layers with 500 units each followed by a ReLU and a dropout layer. This is followed by a 2 unit softmax layer. For MCD, we use the same architecture for the generator except that we use max-pooling instead of convolution layers with stride 2 to downsample the representation. The classifier uses the output of the generator and feeds into a dense layer with 256 units followed by batch normalization and ReLU activation layers. This is followed by a softmax layer.

**Office-31:** For these experiments, we use the publicly available code[2] of the work (Combes et al., 2020) and supply the poisoned data by adding them to the input files being used by the code. We use all default options of the code and use DAN, CDAN, IW-DAN, IW-CDAN algorithms. This is done to eliminate the effect of hyperparameters on the performance of the UDA algorithms on the Office-31 dataset and be able to fairly compare the performance of poisoning.

To obtain the representation trained only on the source domain data, we initialize a ResNet50 model with weight pre-trained on Imagenet. We then update the representation by training on respective source domain data for different tasks.

---

[2]https://bit.ly/34EFb52

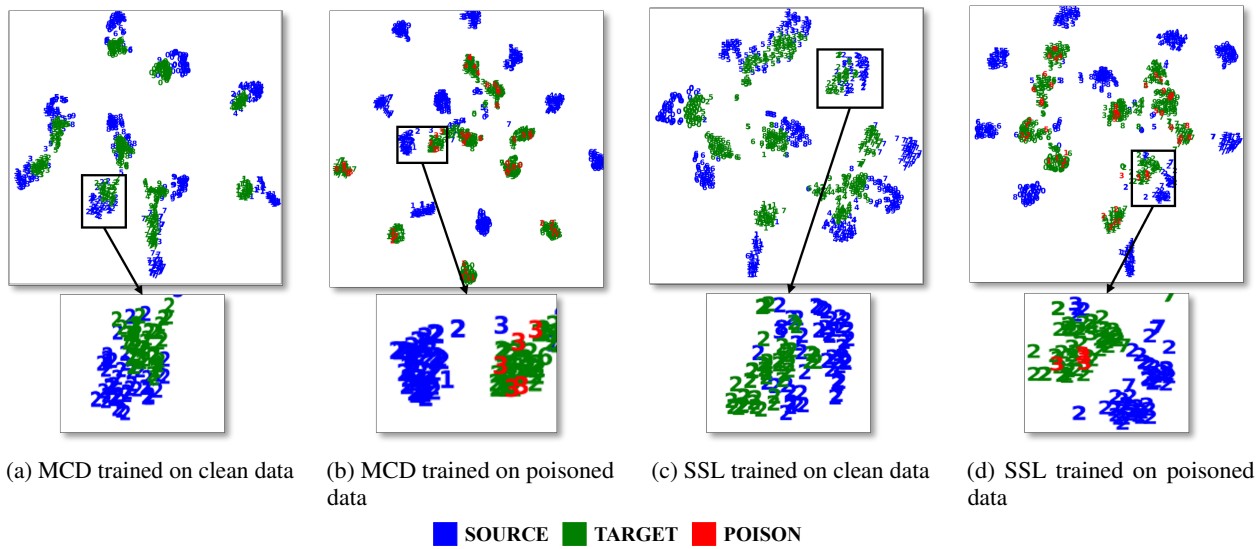

(a) MCD trained on clean data    (b) MCD trained on poisoned data    (c) SSL trained on clean data    (d) SSL trained on poisoned data

■ SOURCE ■ TARGET ■ POISON

*Figure 4.* (Best viewed in color.) t-SNE embedding of the data in the representation space (for MNIST→ USPS task) learnt using MCD and SSL on clean and poisoned source domain data. Without poisoning, correct classes (data from source class 2 is zoomed in) from two domains are aligned ((a) and (c)). The presence of poisoned data prevents the methods from aligning correct classes from the two domains ((b) and (d)).

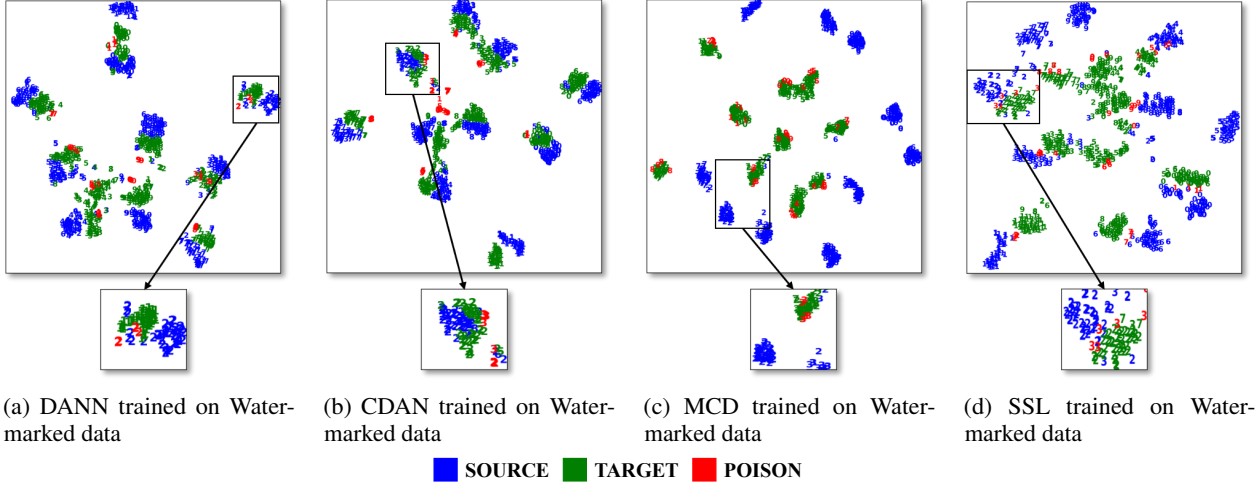

(a) DANN trained on Watermarked data    (b) CDAN trained on Watermarked data    (c) MCD trained on Watermarked data    (d) SSL trained on Watermarked data

■ SOURCE ■ TARGET ■ POISON

*Figure 5.* (Best viewed in color). t-SNE embedding of the data in the representation space (for MNIST→ USPS task) learned using DANN, CDAN, MCD, and SSL poisoned watermarked ($\alpha = 0.3$) data source domain data. Successful poisoning aligns the wrong classes for discriminator-based approaches, as seen in (a) with DANN. Poisoning fails against CDAN because of the pseudo-labels being correct on the target data (b). For MCD, we see 20 distinct clusters highlighting the failure of the method at detecting and aligning target domain data (c). For SSL, the poison data has prevented the correct classes from having very similar representations (d). The failure of most UDA methods with a small amount of watermarked data not only makes our attack practical but also raises serious concerns about the vulnerability of these methods to poisoning attacks.