# OpenReview forum: "On the Effectiveness of Poisoning against Unsupervised Domain Adaptation"
_ICML.cc/2021/Workshop/AML — ICML 2021 Workshop AML Poster_

### Official Review · Reviewer_rttD · 2021-06-19
**Interesting Attacks and Good Theoretical Findings**

**Rating:** Accept
**Confidence:** 3

**Review:**

This paper introduces an upper bound on the test error induced by additive poisoning under the setting where training and test data are from the same distribution. The upper bound explains the difficulty of poisoning deep neural networks using a small amount of poisoned data under this setting. After that, the author explores the poisoning-based attacks in an unsupervised domain adaptation (UDA) setting where the source and the target domain distributions are different.

Pros:
1.	The paper is well-written and the topic is of great significance and is suitable for this workshop.
2.	There are many interesting theoretical and empirical findings in this paper.
3.	The experiments are also extensive whose results are well support author’s claims.

Cons:
1.	I think the author should explicitly illustrate the threat model (e.g., attacker’s capacities and limitations) about the proposed attacks.
2.	I think it would be better if the author can theoretically analyze the effectiveness of poisoning under the UDA setting, even just in a special case. But I understand that it is probably very difficult. I am okay with the current empirical analysis.

---

### Decision · Program_Chairs · 2021-06-21

**Decision:**

Accept (Poster)

**Comment:**

This paper introduced an upper bound on the test error induced by additive poisoning under the setting where training and test data are from the same distribution. The authors can further address the reviewer's comments.